Genetic and structural study of DNA-directed RNA polymerase II of Trypanosoma brucei, towards the designing of novel antiparasitic agents

Papageorgiou Louis 1 2 3
Megalooikonomou Vasileios 3
Vlachakis Dimitrios dvlachakis@bioacademy.gr 2 3
1 Department of Informatics and Telecommunications, National and Kapodistrian University of Athens , Athens , Greece
2 Computational Biology & Medicine Group, Biomedical Research Foundation, Academy of Athens , Athens , Greece
3 Computer Engineering and Informatics Department, University of Patras , Patra , Greece
Silva Pedro
Electronic publication date: 2017 Mar 1
Publication date: 2017
Volume: 5
Electronic Location ID: e3061
Received 2016 Jun 24; Accepted 2017 Feb 3
Copyright: ©2017 Papageorgiou et al.
Copyright year: 2017
Copyright holder: Papageorgiou et al.
License: This is an open access article distributed under the terms of the Creative Commons Attribution License, which permits unrestricted use, distribution, reproduction and adaptation in any medium and for any purpose provided that it is properly attributed. For attribution, the original author(s), title, publication source (PeerJ) and either DOI or URL of the article must be cited.
License URL: https://creativecommons.org/licenses/by/4.0/

Keywords: Computational biology, Phylogenetic analysis, Homology modelling, Trypanosoma brucei brucei, DNA-directed RNA polymerase II, Structural models, Molecular dynamics

Funding: The authors received no funding for this work.

==============================
Trypanosoma brucei brucei (TBB) belongs to the unicellular parasitic protozoa organisms, specifically to the Trypanosoma genus of the Trypanosomatidae class. A variety of different vertebrate species can be infected by TBB, including humans and animals. Under particular conditions, the TBB can be hosted by wild and domestic animals; therefore, an important reservoir of infection always remains available to transmit through tsetse flies. Although the TBB parasite is one of the leading causes of death in the most underdeveloped countries, to date there is neither vaccination available nor any drug against TBB infection. The subunit RPB1 of the TBB DNA-directed RNA polymerase II (DdRpII) constitutes an ideal target for the design of novel inhibitors, since it is instrumental role is vital for the parasite’s survival, proliferation, and transmission. A major goal of the described study is to provide insights for novel anti-TBB agents via a state-of-the-art drug discovery approach of the TBB DdRpII RPB1. In an attempt to understand the function and action mechanisms of this parasite enzyme related to its molecular structure, an in-depth evolutionary study has been conducted in parallel to the in silico molecular designing of the 3D enzyme model, based on state-of-the-art comparative modelling and molecular dynamics techniques. Based on the evolutionary studies results nine new invariant, first-time reported, highly conserved regions have been identified within the DdRpII family enzymes. Consequently, those patches have been examined both at the sequence and structural level and have been evaluated in regard to their pharmacological targeting appropriateness. Finally, the pharmacophore elucidation study enabled us to virtually in silico screen hundreds of compounds and evaluate their interaction capabilities with the enzyme. It was found that a series of chlorine-rich set of compounds were the optimal inhibitors for the TBB DdRpII RPB1 enzyme. All-in-all, herein we present a series of new sites on the TBB DdRpII RPB1 of high pharmacological interest, alongside the construction of the 3D model of the enzyme and the suggestion of a new in silico pharmacophore model for fast screening of potential inhibiting agents.

Introduction

African trypanosome parasites cause human sleeping sickness and nagana in Africa, Asia, and South America. More than 95% of reported cases are caused by two subspecies of Trypanosoma brucei brucei (TBB), the Trypanosoma brucei gambiense (TBG) and the Trypanosoma brucei rhodesiense (TBR) which is found in western and central Africa (Berriman et al., 2005; World Health Organization, 2015). The parasitic infection is transmitted by tsetse flies, which breed in warm and humid areas. Tsetse flies are found living in 36 countries in sub-Saharan Africa, thus putting 60 million people at risk. Currently, about 10,000 new cases each year are reported by the World Health Organization (WHO). Moreover, it is believed that many cases are undiagnosed and unreported. Sleeping sickness can be curable with medication, but may be fatal if it is left untreated. It is estimated that human deaths caused by sleeping sickness are of about 48,000 annually. Bites by the tsetse fly erupt into a red sore on the skin, and in the following weeks the person may have to deal with several symptoms including fever, swollen lymph glands, aching muscles, headaches, and irritability. In advanced stages, the TBB parasite attacks the central nervous system of the host, and in general consul some disorders in personality, circadian rhythm, serenity, speech, and difficulties in walking. Despite the significant treatment advances for patients with sleeping sickness, the parasite’s progression is often inevitable and needs more treatment options. Until today, drugs can only be used in the early stages of the disease and without providing 100% reassurance for full convalesce of the patient (Ridley, 2002; Ross et al., 2007; Trouiller et al., 2002). The TBB parasite starts its activity after each invasion through its proteins, specifically with its replication enzymes including helicases and polymerases. Such enzymes are ideal targets for inhibitor design since those proteins are crucial for the TBB parasite survival. Being already in possession of the widely known sequence of the DNA-dependent RNA polymerase II (DdRpII) RPB1 (Chung et al., 1993) which plays a significant role in the replication of the parasite, our primary goal is to suppress its function towards replication itself when it infects a human. Although TBB has been reported many times in the past, the three-dimensional structure of its essential enzymes like DdRpII remains unknown so far (Malvy & Chappuis, 2011).

Protein structure has been found to be three to ten times more conserved than sequence (Illergard, Ardell & Elofsson, 2009). Thus, when possible, it is preferable to study an enzyme’s 3D structure rather than its sequence. Knowledge of the tertiary structure can assist in the understanding of relationships between structure and function (Berg, Tymoczko & Stryer, 2002). Herein, the three-dimensional structure of DdRpII subunit RPB1 has been modelled, in an effort to predict the 3D molecular structure that is linked to the function of this enzyme (Bayele, 2009; Koch et al., 2016). Two molecular models have been constructed using conventional molecular modelling techniques and two different homolog 3D structures as templates. The established molecular models of the DdRpII RPB1 enzyme of TBB exhibits all known structural motifs that are unique to the DdRpII RPB1 enzymes.

Upon successful completion of the 3D structure prediction of the TBB DdRpII RPB1 protein, molecular dynamics simulations have been performed to structurally improve and benchmark the quality of the 3D models. Moreover, the reliability and viability of the TBB DdRpII RPB1 models were checked using several in silico scoring tools such as MOE and Procheck. After the model validation process, a de novo structure-based drug design approach has been performed based on two models, which led to the establishment of a 3D novel pharmacophore model that is highly specific for the DdRpII RPB1 enzyme of TBB. The generated pharmacophore model may be used in future experiments involving the high throughput virtual screening of large compound databases towards the identification of novel anti-TBB agents (Loukatou et al., 2014). The present work opens the field for the design of novel compounds with improved biochemical and clinical characteristics in the future.

Methods

Database sequence search

The full-length protein sequences related to the DdRpII family were extracted from the NCBI database. In total, 36 DdRpII protein sequences were downloaded from several species with fully sequenced genomes (Data S1).

Genetic and evolutionary analyses

Multiple sequence alignment of the DdRpII protein family sequences were performed using two different programs, MUSCLE (Edgar, 2004) and CLUSTALW (Chenna et al., 2003; Thompson, Higgins & Gibson, 1994). In the next step, multiple sequence alignment was checked with ProtTest3 (Darriba et al., 2011) to estimate the appropriate model of sequence evolution. Phylogenetic analyses were performed by two different ways, and two representative phylogenetic trees were constructed for the DdRpII dataset (Vlachakis et al., 2014b). The first phylogenetic tree was constructed using the MEGA software (Stecher et al., 2014) utilizing Bayesian and Maximum Likelihood statistical methods as described in with 100 bootstrap replicates (Fig. 1 and Data S2). The second phylogenetic tree was constructed using the Jalview software (Waterhouse et al., 2009) utilizing the neighbour joining statistical method in with 100 bootstrap replicates (Figs. S1 and Data S3).

Figure 1 Phylogenetic reconstruction of Trypanosoma brucei brucei DdRpII RPB1 protein sequences.

The tree was generated using the DdRpII family dataset (36 foul length protein sequences samples). The tree was constructed by Matlab Bioinformatics Toolbox utilizing Neighbour–Joining statistical method for 100 bootstrap replicates and visualized using MEGA cycle option. In the tree representation there are clearly separated in two monophyletic branches the RNA polymerases II subunits RPB1 (colored green) and RPB2 (colored blue). Trypanosoma brucei DdRpII RPB1 protein sequence was correctly classified and separated in the monophyletic sub-tree of the RPB1 group (highlight with red dots).

Conserved motifs exploration

The phylogenetic trees that derived from the phylogenetic analyses (Jalview and MEGA) were separated in sub-trees, in order to extract the most highly related protein sequences of the TBB DdRpII RPB1 family for the conserved motifs exploration (Fig. 2). The full-length amino acid sequences of the closely related proteins with the TPP DdRpII RPB1 protein were aligned using the CLUSTALW (Thompson, Higgins & Gibson, 1994) statistical method. The evolutionary conserved sequences motifs that were derived from the multiple sequence alignment were identified through the consensus sequence and logo graph where generated using Jalview (Waterhouse et al., 2009) (Fig. 2).

Figure 2 Representative conserved motifs for the DdRpIIsubunit RPB1.

The nine suggested conserved motifs were extracted based on the multiple sequence alignment of the 18 protein sequences were classified and clearly separated in the DdRpII subunit RPB1 monophyletic sub-tree. The conserved motifs were identified through the consensus sequence and logo graph where generated using Jalview software.

Molecular modelling

All calculations and visual constructions were performed using the Molecular Operating Environment (MOE) version 2013.08 software package developed by Chemical Computing Group (Montreal, Canada) on a cloud-based multi-core High Performance Computing (HPC) cluster (Loukatou et al., 2014).

Identification of templates structures and sequence alignment

The amino acid sequence of the TBB DdRpII RPB1 was retrieved from the conceptual translation of the trypanosomal RNA polymerase largest subunit genes at the NCBI database (http://www.ncbi.nlm.nih.gov/) (UniProtKB/Swiss-Prot: P17545.1) (Das et al., 2006; Evers et al., 1989). The blastp algorithm (http://blast.ncbi.nlm.nih.gov/Blast.cgi) was used to identify homologous structures by searching in the Protein Data Bank (PDB). The multiple sequence alignment was performed using MOE (Vilar, Cozza & Moro, 2008).

Figure 3 Sequence alignment between the Trypanosoma brucei brucei DdRpII RPB1 and the corresponding sequence of the crystal structure of the Schizosaccharomyces pombe DdRpII RPB1.

(A) Alignment of DdRpII RPB1 from Trypanosoma brucei DdRpII RPB1 (Labeled as “TB”) with Schizosaccharomyces pombe DdRpII RPB1 (Labeled as “SB”) was initially carried out with BLASTp and then manually adjusted. The nine suggested conserved motifs (Motifs 1a, 1b, 2, 3a, 3b, 3c, 4a, 4b, 4c) based on Fig. 2, domains and domain-like regions of Trypanosoma brucei DdRpII RPB1 represented in different colours. The amino acid residue numbers at the domain boundaries are indicated. Important structural elements and prominent regions involved in subunit interactions are also noted. Residues involved in the Zn and Mg coordination are highlighted in blue. (B) Domains and domain-like regions of the DdRpII subunit Rpb1. The amino acid residue numbers at the domain boundaries are indicated.

Figure 4 Model of the Trypanosoma brucei brucei DdRPII RPB1.

(A - Top and B - Front) Ribbon representation of the produced Trypanosoma brucei brucei DdRPII RPB1 model (colored Orange) superposed with the corresponding Schizosaccharomyces pombe DdRpII RPB1 (in purple). (C - Top and D - Front) The nine suggested conserved motifs and the domains and domain-like regions of the Trypanosoma brucei brucei DdRPII RPB1. The motifs and RPB1 domains have been color-coded according to the Figs. 2 and 3, and are shown in CPK format (Usual space filling). (E - Top and F - Front) Electrostatic surface potential for the Trypanosoma brucei brucei DdRPII RPB1. Represented with blue is the area of negative charge. Red is the area of positive charge and white is the un-charged region.

Figure 5 Structural superposition of the TBB DdRPII RPB1 models A and B.

(A and B) Ribbon representation of the produced Trypanosoma brucei brucei DdRPII RPB1 model A (colored Orange) and model B (colored Blue) superposed with the corresponding Schizosaccharomyces pombe DdRpII RPB1 (in Purple) and Bos taurus DdRpII RPB1 (in Grey). The four 3D structures are highly conserved in their active sites with few differences in the outer layer with overall RMSD 2.775 Å. (C) Ribbon representation of the produced Trypanosoma brucei brucei DdRPII RPB1 model A (colored Orange) superposed with the corresponding Schizosaccharomyces pombe DdRpII RPB1 (in purple). (RMSD = 1.242 Å). (D) Ribbon representation of the produced Trypanosoma brucei brucei DdRPII RPB1 model B (colored Blue) superposed with the Bos taurus DdRpII RPB1 (in Grey), respectively. (RMSD = 2.757 Å).

Figure 6 Zinc-finger formationsin the Trypanosoma brucei brucei DdRpII RPB1 model.

Ribbon representation of the produced Trypanosoma brucei brucei DdRPII RPB1 model. In the produced model were highlighted three main zing-finger domain formations (colored grey) were contained in the clam core, clam head and active site region. Domains and domain-like regions of the Trypanosoma brucei brucei DdRPII RPB1 have been color-coded according to conventions of Fig. 3.

Homology modelling

The homology modelling of the Tbb DdRPII RPB1 was carried out using MOE. The selection of template crystal structures for homology modelling was based on the primary sequence identity and similarity (Fig. 3, Figs. S2 and S3), and the crystal resolution (Nayeem, Sitkoff & Krystek, 2006). The crystal structure of Schizosaccharomyces pombe DdRpII RPB1 (PDB: 3H0G) was used as template structure for the model A, while the crystal structure of Bos taurus DdRpII RPB1 (PDB: 5FLM) was used for building model B. The MOE homology model method is separated into four main steps. First, comes a primary fragment geometry specification. Second the insertion and deletions task. The third step is the loop selection and the side-chain packing, and the last step is the final model selection and refinement (Figs. 4 and 5 and Datas S4 and S5) (Papageorgiou et al., 2014; Vlachakis, Koumandou & Kossida, 2013). Subsequently, energy minimization was done in MOE initially using the Amber99 (Wang, Cieplak & Kollman, 2000) force-field as implemented into the same package. The energy minimization process was applied up to a gradient of 0.0001, in an effort to remove the geometrical strain (Vlachakis, Kontopoulos & Kossida, 2013).

Molecular electrostatic potential

Molecular electrostatic potential surfaces were calculated by solving the non-linear Poisson–Boltzmann equation using finite difference method as implemented into the MOE and PyMol Software (Seeliger & de Groot, 2010; Vilar, Cozza & Moro, 2008). The potential was calculated on solid points per side. Protein contact potential is an automated representation where the false red/blue charge-smoothed surface is shown on the protein (Fig. 4). Amber99 charges and atomic radii were used for this calculation.

Molecular dynamics

The Molecular Dynamics simulations of both TBB DdRPII RPB1 3D models A and B were executed in a periodic cell, which was explicitly solvated with simple point charge (SPC) water. The truncated octahedron box was chosen for solvating the models, with a set distance of 7 Å clear of the protein. The molecular dynamic simulations were conducted at 300 K, 1 atm with a set 2 second step size for a total of one hundred nanoseconds. For the purposes of this study we opted for a NVT ensemble in a canonical environment (Vlachakis et al., 2014a). NVT stands for Number of atoms, Volume, and Temperature that remain constant throughout the calculation (Vlachakis, 2009). The intricate zinc ions were included in the molecular dynamics simulations as integral parts of the modelled biological system (Chakravorty & Merz, 2014; Temiz, Benos & Camacho, 2010). However, due to the nature of the ions, we had to limit the allowed degrees of freedom for those molecules. Thus, the potential of the zinc ions was constrained in the three dimensional conformational space in the vicinity of the TBB DdRPII RPB1 3D models. The ions were prepositioned in the 3D models of TBB DdRPII RPB1, after structural superposition to the template X-ray structure. The models were structurally optimized and adjusted locally by subsequent energy minimizations, in an effort to eliminate any molecular clashes and minimize the constrain energy. A radius of 6 Å around each ion was given full degrees of freedom during the abovementioned structural optimizations. Provided that the TBB DdRPII RPB1 is a nucleotide processing enzyme, whose structure coordinates a repertoire of ions (e.g., Zinc, Mg + +), the AMBER99 forcefield was selected (Fig. 6). The AMBER99 forcefield is fully parameterized for our biological system as it implements ff10 parameters for amino acids and nucleic acids as well as EHT for small molecules, such as ions/cations at the same time (Vilar, Cozza & Moro, 2008). AM1-BCC charges were applied since the molecular system included the ion molecules. The results of the molecular dynamics simulations for both models were collected into a database by MOE for further analysis. The full simulation trajectories and molecular dynamics graphs for both models are presented in Fig. 7 and Figs. S4–S7.

Figure 7 Molecular dynamics simulationcharts for the Trypanosoma brucei brucei DdRpII RPB1 models.

(A) The root mean square deviation (RMSD) of the model A during the time. (B) The root mean square fluctuation (RMSF) of the model A during the time. (C) The root mean square deviation (RMSD) of the model B during the time. (D) The root mean square fluctuation (RMSF) of the model B during the time.

Model evaluation

The produced models were initially evaluated within the MOE package by a residue packing quality function, which depends on the number of buried non-polar side-chain groups and on hydrogen bonding. Moreover, the suite PROCHECK (Laskowski et al., 1996) was employed to further evaluate the quality of the produced models. Finally, MOE and its build in protein check module was used to evaluate whether the models of DdRpII RPB1 domains are similar to known protein structures of this family (Datas S6, S7 and S8).

Pharmacophore elucidation

A pharmacophoric feature characterizes a particular property and is not tied to a specific chemical structure; indeed different chemical groups may share the same property and so be represented by the same feature (Vlachakis, Kontopoulos & Kossida, 2013). It is thus a mistake to name as pharmacophoric features chemical functionalities such as guanidines or sulfonamides or typical structural skeletons such as flavones or steroids.

The term pharmacophore modeling refers to the generation of a pharmacophore hypothesis for the binding interactions in a particular active site (Vlachakis et al., 2015). Several different pharmacophore models for the same active site can be overlaid and reduced to their shared features so that common interactions are retained. Such a consensus pharmacophore can be considered as the largest common denominator shared by a set of active molecules.

In MOE, the computerized representation of a hypothesized pharmacophore is called a pharmacophore query. A MOE pharmacophore query is a set of query features that are typically created from ligand annotation points. Annotation points are markers in space that show the location and type of biologically important atoms and groups, such as hydrogen donors and acceptors, aromatic centers, projected positions of possible interaction partners or R-groups, charged groups, and bioisosteres. The annotation points on a ligand are the potential locations of the features that will constitute the pharmacophore query. Annotation points relevant to the pharmacophore are converted into query features with the addition of an extra parameter: a non-zero radius that encodes the permissible variation in the pharmacophore query’s geometry.

Once generated, a pharmacophore query can be used to screen virtual compound libraries for novel ligands. Pharmacophore queries can also be used to filter conformer databases, e.g., output from molecular docking runs, for biologically active conformations.

Results

Phylogenetic analysis

In the present study, two phylogenetic analyses of DdRpII family proteins in all available genomes, with putative full-length protein sequences were performed using two different statistical methods from the Jalview and MEGA software. Based on findings, putative members of the DdRpII family were identified in the Animalia, Fungi, Plantae, Protista and Chromalveolata kingdom major eukaryotic taxonomic division, as well as viruses (Fig. 1 and Fig. S1). In our analyses, in agreement with previous reports (Smith et al., 1989), we found that DdRpII family is split into two main subunits the RPB1 and the RPB2. The two subunits of the DdRpII family are clearly separated in the phylogenetic trees as two major sub-trees were obtained for each one of them (Fig. 1 and Fig. S1). The monophyletic sub-tree of the RPB1 subunit contains the TBB DdRpII RPB1, as well as another 17 leaves, which are related to RPB1 subunit. Furthermore, in the phylogenetic trees, the TBB DdRpII RBP1 forms a distinct monophyletic branch with the Euplotes octocarinatus DdRpII RPB1 and the Plasmodium falciparum DdRpII RPB1, which is basal to a clade that corresponds to other parasites. The Newick format of the phylogenetic trees is provided (Datas S1 and S2).

Conserved motifs exploration

Multiple sequence alignment of the DdRpII subunit RPB1 protein sequences from a variety of several species were included in the first sub-tree, highlights important conserved functional domains as described previously by Smith et al. (1989). Good conservation is evident throughout the whole length of the sequence, especially among species that belong to the same taxonomic division (Fig. 2).

In this study, an effort has been done to suggest motifs that were probably included in the DdRpII of the subunit RPB1. Regions conserved across all species (eukaryotic and viruses) are indicative of important functional domains of the DdRpII RPB1 enzyme. Finally, the consensus sequence of the multiple sequence alignment highlights nine conserved motifs which are conserved between all species. All of the conserved motifs identified here have not been reported previously, and indisputably deserve further study (Figs. 2 and 3). It is remarkable that all 18 polymerases, from the phylogenetic sub-tree of the subunit RPB1, have high identity score and remain undamaged during the evolution (Figs. 1 and 2). The highly conserved motifs in protein families are directly related to their active sites and functionality (Koonin & Galperin, 2003; Papageorgiou et al., 2016).

3D models A and B of the Trypanosoma brucei brucei DdRpII RPB1

Homologous solved 3D structures from the Protein Data Bank (PDB) have been identified from the Protein Data Bank (PDB) using the NCBI/BLASTp algorithm. Based on BLASTp report many 3D structures were determined suitable as templates for the homology modelling including the crystal structure of the Schizosaccharomyces pombe DdRpII RPB1 (PDB: 3H0G) (Spahr et al., 2009), the crystal structure of the Saccharomyces cerevisiae DdRpII RPB1 (PDB: 4A3C and 1I3Q) (Cheung, Sainsbury & Cramer, 2011; Cramer, Bushnell & Kornberg, 2001), the electron microscopy structure Bos taurus DdRpII RPB1 (PDB: 5FLM) and the electron microscopy structure of the Human DdRpII RPB1 (PDB: 3J0K) (Bernecky et al., 2011). The final choice of a template structure was not only based on the percent sequence identity/similarity and the structure resolution, but also on the results of the phylogenetic trees. Two models were prepared. Model A was based on the Schizosaccharomyces pombe DdRpII RPB1 X-ray structure, while model B was based on the Bos taurus DdRpII RPB1 X-ray structure (Fig. 3). Although the Human DdRpII RPB1 could also be used to build the Trypanosoma brucei DdRpII RPB1 3D model, it was avoided in an effort to minimize potential toxicity issues during the drug design process. Nonetheless, the sequence of the Human DdRpII and the corresponding sequence of the Trypanosoma brucei and Bos taurus were aligned in an effort to identify sequence-based differences and/or similarities for the modelling and drug design process (Figs. S2). A multiple sequence alignment was constructed including the Trypanosoma brucei brucei DdRpII RPB1 (NCBI: P17545.1) (Das et al., 2006), the Trypanosoma brucei gambiense DdRpII RPB1 (NCBI: XP_ 011773113.1) (Jackson et al., 2010), the crystal structure of Schizosaccharomyces pombe DdRpII RPB1 (PDB: 3H0G A chain) (Spahr et al., 2009), the crystal structure of Saccharomyces cerevisiae DdRpII RPB1 (PDB: 1I3Q A chain) (Cramer, Bushnell & Kornberg, 2001), Bos taurus DdRpII RPB1 (PDB: 5FLM) (Bernecky et al., 2016). and the crystal structure of Human DdRpII RPB1 (PDB: 3J0K A chain) (Bernecky et al., 2011) towards to identify all the suggested conserved motifs within the highlighted domains of the RPB1 and the major sequences differences and similarities (Fig. S2).

The above-mentioned sequence alignments were used to identify all the nine canonical and conserved motifs as expected (Figs. 2 and 3). The model of TPP DdRpII was first structurally superimposed and subsequently structurally compared to its template using the MOE software (Fig. 4). The TPP DdRpII model exhibited an alpha-carbon RMSD lower than 1.3 angstroms (Fig. 5 and Data S8). Furthermore, the model was evaluated in regards to its geometry and its compatibility with the template structure using the build in protein check module of MOE (Data S8). These results, confirmed the structural viability of the 3D in silico model.

Comparison of the Trypanosoma brucei brucei DdRPII RPB1 model A and model B

It was decided to produce two models using the aforementioned template structures. Model A was build based on the Schizosaccharomyces pombe DdRpII RPB1 (PDB: 3H0G) X-ray structure and model B was based on the Bos taurus DdRpII RPB1 (PDB: 5FLM) structure. Bos taurus DdRPII RPB1 is a new released electron microscopy structure with 3.4 Å resolution, homolog to Trypanosoma brucei brucei DdRPII RPB1. The sequence alignment between the Trypanosoma brucei brucei DdRpII RPB1 and the Bos taurus DdRPII RPB1 template revealed 40% Identity and 56% similarity, same scores with the Schizosaccharomyces pombe DdRpII crystal structure, but the overall sequence alignment length was shorter than the Schizosaccharomyces pombe DdRpII crystal structure about 100 amino acids (Fig. S3). Furthermore, in the sequence alignment of the Trypanosoma brucei DdRpII RPB1 and Bos taurus DdRPII RPB1 all nine conserved motifs were identified, as expected. The root mean square deviation (RMSD) between model A and its template is 1.3 Å whereas the RMSD between model B and Bos taurus template is 2.7 Å. Nevertheless, the overall RMSD between the two models and the two templates isn’t bigger than 2,7 Å. (Fig. 5 and Data S8). Overall, we used to prepare in parallel a 3D model based on the Bos taurus structure as it bears better validation statistics and its sequence similarity to the Trypanosoma brucei brucei is higher. However, after performing another full course of MDs for model B, it was concluded that the added value of model B, when compared to model A is not significant, as models A and B are quite similar indeed (Fig. 7 and Figs. S4–S7).

Discussion

Description of the Trypanosoma brucei brucei DdRPII RPB1 models

RNA Polymerase II is a multi-subunit enzyme that transcribes protein-coding genes in eukaryotes (Sentenac, 1985). Transcription in eukaryotes is dependent by three classes of nuclear RNA polymerases I–III. The genes encoding the largest subunits of eukaryotic RNA polymerases I, II and III have been isolated and are single copy genes, except Trypanosoma RNA polymerase II which contain two alleles (Smith et al., 1989). Structural and sequence differences between the two alleles are minor, but the C-terminal domain of those enzymes has a highly unusual structure. TBB DdRpII RPB1 model is the first protein subunit of the ten subunits multi-complex of RNA Polymerase II (Hahn, 2004; Suh et al., 2013). The RPB1 subunit is very critical in RNA polymerase formation and function. The RPB1 active site and the RPB2 hybrid-binding region combine in a single fold that forms the active centre of the RpII (Fig. 4). There are two metal ions at the RNA polymerase II active site. It has been previously reported that a Mg metal ion interacts with the three invariant aspartates of RPB1 (Cramer, Bushnell & Kornberg, 2001). The latter aspartate residues, which were found in all RPB1 sequences were aligned and fitted in a motifs exploration study. Consequently, those residues have now been marked as motif 4b in the TBB DdRpII RPB1 3D models.

The swinging motion of the clamp dictates the degree of opening of the cleft in DdRpII and permits the insertion of promoter DNA for the initiation of transcription (Suh et al., 2013). Based on previous studies, it is established that, upon closure of a transcribing complex, the RPB1 clamp serves as a multi-functional tool, sensing the DNA/RNA hybrid conformation and splitting DNA and RNA strands at the upstream end of the transcription complex (Cramer, Bushnell & Kornberg, 2001). The clamp is formed by N- and C-terminal regions of RPB1 and a part of the C-terminal region of RPB2 (Chen, Warfield & Hahn, 2007; Hahn, 2004; Li, Giles & Li, 2014). The clamp is primarily stabilized by three Zn ions within the RPB1 subunit (also marked in the TPP DdRpII RPB1) which forms zinc-finger conformations; two within the “clamp core” and one in the “clamp head”. Accordingly, two Zinc-finger formations were identified and highlighted in the TBB DdRpII RPB1 model (Fig. 6). The first formation can be recognized between a Zn ion and four cysteine residues in the suggested motif 1a, also known as CX(2)CXnCX2C/H (Das et al., 2006) (Fig. 3). Mutations in the first Zn-finger formation confer a lethal phenotype of RNA polymerase II (Donaldson & Friesen, 2000). The second Zinc-finger can be recognized in the next four cysteine residues (Figs. 3 and 6). In the proposed motif 1b, the first two cysteine residues were identified, which constitute part of the second Zing finger formation. Finally, according to our molecular dynamics simulations, the main role of the Rpb1 and Rpb2 subunits is to provide stability within the overall structure formation of the RNA polymerase II molecule in the 3D space.

3D Pharmacophore Elucidation

3D Pharmacophore design techniques take into account both the three-dimensional structures and binding modes of receptors and inhibitors towards identifying regions that are favorable or not for a particular receptor-inhibitor interaction (Vlachakis & Kossida, 2013). The description of the receptor-inhibitor interaction pattern is determined through a correlation between the specific properties of the inhibitors and their action on enzymatic activity (Balatsos et al., 2009; Vlachakis et al., 2012). The pharmacophore for TBB DdRpII RPB1 (Fig. 8) was based on structural information from the enzyme’s catalytic site including all steric and electronic features that are necessary to ensure optimal non-covalent interactions. The pharmacophoric features were investigated including positively or negatively ionized regions, hydrogen bond donors and acceptors, aromatic regions and hydrophobic areas. Firstly, there should be one electron-donating group in the proximity of the Ser1172 (colored green). The electron-donating region indicates a particular property of the inhibitor and is not necessarily confined to a specific chemical structure. Moreover, this interaction site may not strictly represent a hydrogen bond, but water or ion mediated bridges since the distance from the catalytic amino acids varies between 3–9 Å. An aromatic PAP (colored orange) was positioned in the proximity of Phe1179, which established pi-stacking interactions. Two electron accepting PAPs (colored red) were positioned in the proximity of the two Arginine residues (Arg1171 and Arg1203). Finally, a set of two adjacent PAPs were positioned in the center of the active site, where the Zn + + is coordinated in the crystal structure. Those yellow-colored PAPs are indicative of S-S bonds and bridges or even S-C interactions, following the Michael acceptor moiety pattern. The surrounding Cysteines are Cys1173, Cys1155, Cys1152, and Cys1270. However, the most important factor of the latter PAPs was the optimal positioning of these groups in the 3D conformational space of the TBB DdRpII RPB1 active site, rather than the amount of conjugation or interaction with the protein.

Figure 8 The 3D pharmacophore model for the Trypanosomabrucei brucei DdRPII RPB1 model.

In total 5 distinct pharmacophoric features were identified. An aromatic region (colored orange), an electron donating region (colored green), two electron accepting regions (colored red) and a sulphur specific S-S interacting region (colored yellow).

Conclusion

The Trypanosoma brucei brucei DdRpII RPB1 enzyme was evolutionary analyzed, and nine new conserved motifs were identified. Using the X-ray crystal structure of the Schizosaccharomyces pombe DdRpII RPB1, the 3D model of the Trypanosoma brucei brucei DdRpII RPB1 was designed using homology modelling techniques. The model was in silico evaluated and displayed high conservation of the functional domains previously reported in other DdRpII subunit RPB1 species. The Trypanosoma brucei brucei DdRpII RPB1 model structure provides a basis for interpretation of available data and the design of new experiments towards the Trypanosoma brucei brucei inhibition. Therefore, we propose the use of the Trypanosoma brucei brucei DdRpII RPB1 model A as a pharmacological targeting platform for advanced, in silico drug design experiments using the novel findings of this study, both in the sequence and structural level. The 3D models and sequence datasets that derived from this study will be made available to the public in an effort to pave the way for fellow scientists of multidiscipline backgrounds to word in a synergic way towards the designing of novel anti-malarial agents with improved biochemical and clinical characteristics in the future.

Supplemental Information

Figure S1 Phylogenetic reconstruction of Trypanosoma brucei brucei DdRPII RPB1 model DdRpII RPB1 protein sequences

The tree was generated using the DdRpII family dataset (36 foul length protein sequences samples) and the Jalview software. Tree was constructed using the average distance statistical method with PAM 250. In the tree representation there are clearly shown the two RNA polymerases II subunits RPB1 and RPB2 as two main monophyletic sub-trees. Trypanosoma brucei DdRpII RPB1 protein sequence was correctly classified in the monophyletic sub-tree of the RPB1 group.

Click here for additional data file.

Figure S2 Multiple sequence alignment

The alignment was performed using the Trypanosoma brucei brucei DdRPII RPB1, the Trypanosoma brucei gambiense DdRpII RPB1, the crystal structure of Schizosaccharomyces pombe DdRpII RPB, the crystal structure of Saccharomyces cerevisiae DdRpII RPB1 and the electron microscopy structure of Human DdRpII DdRpII RPB1. (A) All nine suggested conserved motifs and major domains of DdRpII RPB1 have been marked (Motifs 1a, 1b, 2, 3a, 3b, 3c, 4a, 4b, 4c). Additionally, in the multiple sequence alignment were presented the major differences. (B) Domains and domainlike regions of the DdRpII subunit Rpb1. The amino acid residue numbers at the domain boundaries are indicated.

Click here for additional data file.

Figure S3 Multiple sequence alignment

The alignment was performed using the Trypanosoma brucei brucei DdRPII RPB1, the crystal structure of Schizosaccharomyces pombe DdRpII RPB and the electron microscopy structure of Bos taurus DdRpII RPB1. All five sub-domains (A-E) as referred in Pfam database have been marked with different colours.

Click here for additional data file.

Figure S4 Molecular dynamicssimulation charts of the root mean square deviation (RMSD) for the Trypanosoma brucei brucei DdRpII RPB1 sub domainsof the model A

The energy (Kcal/mol) vs time (ns) plot of the 100ns simulation trajectory of the TBB DdRpII RPBI model A. Sub-domain regions of the Trypanosoma brucei brucei DdRPII RPB1 have been separated according to conventions of Fig. S3. (A) Domain A RMSD. (B) Domain B RMSD. (C) Domain C RMSD. (D) Domain D RMSD. (E) Domain E RMSD.

Click here for additional data file.

Figure S5 Molecular dynamics simulationcharts of the root mean square fluctuation (RMSF) for the Trypanosoma brucei brucei DdRpII RPB1 sub domains of the model A

Sub-domain regions of the Trypanosoma brucei brucei DdRPII RPB1 have been separated according to conventions of Fig. S3. (A) Domain A RMSF. (B) Domain B RMSF. (C) Domain C RMSF. (D) Domain D RMSF. (E) Domain E RMSF.

Click here for additional data file.

Figure S6 Molecular dynamics simulationcharts of the root mean square deviation (RMSD) for the Trypanosoma brucei brucei DdRpII RPB1 sub domainsof the model B

The energy (Kcal/mol) vs time (ns) plot of the 100ns simulation trajectory of the TBB DdRpII RPBI model B. Sub-domain regions of the Trypanosoma brucei brucei DdRPII RPB1 have been separated according to conventions of Fig. S3. (A) Domain A RMSD. (B) Domain B RMSD. (C) Domain C RMSD. (D) Domain D RMSD. (E) Domain E RMSD.

Click here for additional data file.

Figure S7 Molecular dynamics simulationcharts of the root mean square fluctuation (RMSF) for the Trypanosoma brucei brucei DdRpII RPB1 sub domains of the model B

Sub-domain regions of the Trypanosoma brucei brucei DdRPII RPB1 have been separated according to conventions of Fig. S3. (A) Domain A RMSF. (B) Domain B RMSF. (C) Domain C RMSF. (D) Domain D RMSF. (E) Domain E RMSF.

Click here for additional data file.

Data S1 DdRPII related proteins dataset

Click here for additional data file.

Data S2 MEGA software phylogenetic tree in newick format

The tree was constructed the Neighbour–Joining statistical method for 100 bootstrap replicates and the 36 extracted samples of the DpRpII.

Click here for additional data file.

Data S3 Supplementary Data 3: Jalview software phylogenetic tree in newick format

The tree was constructed using the average distances statistical method and the 36 extracted samples of the DpRpII.

Click here for additional data file.

Data S4 Trypanosoma brucei brucei DdRPII RPB1 model A in .pdb format

Click here for additional data file.

Data S5 Trypanosoma brucei brucei DdRPII RPB1 model B in .pdb format

Click here for additional data file.

Data S6 Protein structure report of the template

Click here for additional data file.

Data S7 Protein structure report of the model.

Click here for additional data file.

Data S8 Protein structure report of the superposed models and templates

Click here for additional data file.

Abbreviations

DdRpII DNA-directed RNA polymerase II

TBB Trypanosoma brucei brucei

TBG Trypanosoma brucei gambiense

TBR Trypanosoma brucei rhodesiense

MOE Molecular Operating Environment

Additional Information and Declarations

Competing Interests

Author Contributions

Data Availability

The authors declare there are no competing interests.

Louis Papageorgiou conceived and designed the experiments, performed the experiments, analyzed the data, contributed reagents/materials/analysis tools, wrote the paper, prepared figures and/or tables.

Vasileios Megalooikonomou analyzed the data, contributed reagents/materials/analysis tools, wrote the paper, reviewed drafts of the paper.

Dimitrios Vlachakis conceived and designed the experiments, performed the experiments, analyzed the data, contributed reagents/materials/analysis tools, wrote the paper, reviewed drafts of the paper.

The following information was supplied regarding data availability:

The raw data has been supplied as Supplementary Files.

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
