# Peer review of "Genetic and structural study of DNA-directed RNA polymerase II of Trypanosoma brucei, towards the designing of novel antiparasitic agents"

_PeerJ, doi:10.7717/peerj.3061_

## Round 0.1 · original submission · Major Revisions

Although the reviews are positive, there are still several important portions in your manuscript which require longer discussion. The reviewers point out that sparse details regarding your MD simulation are provided, and I don't understand the lack of graphs/data arising from such a relatively long and expensive 100 ns simulation. At a minimum, I would expect to find RMSD graphs of the overall structure and of each domain and an analysis of the pharmacophore pockets along the simulation (i.e. whether they remain accessible to solvent/open, etc.).

Personal observations from the editor:

You fail to provide hard data regarding the quality of your protein model. You state the values of Verify3D scores, but you do not tell the reader what range of scores are expected from a good model. Ramachandran plots, clashes, etc. are also missing. A quick run of your model through MolProbity (http://molprobity.biochem.duke.edu/) showed a very poor Ramachandran plot: only 76% of the residues fall in the most favored region whereas >98% are expected from good models., and you get a large number of residues in forbidden regions (>5%). Does the quality of the structure improve upon MD simulation?
You may checkthe molprobity results in the following two links:

http://molprobity.biochem.duke.edu/data/7ttmnn5ev6afkm92ggkvnjmvq3/charts/peerj-11326-Supplementary_data_3_clean-rama.pdf

http://molprobity.biochem.duke.edu/viewtable.php?MolProbSID=7ttmnn5ev6afkm92ggkvnjmvq3&file=/home/rlab/Desktop/MolProbities/MolProbity160425_4.3original/public_html/data/7ttmnn5ev6afkm92ggkvnjmvq3/raw_data/peerj-11326-Supplementary_data_3_clean-multi.table

The molecular descriptors from the pharmacophore-3D evaluation should also be presented with more detail.

Reviewer 1 ·

Basic reporting

The authors have carried out bioinformatics and molecular modeling studies in order to built a structural model of the enzyme TB DdRpII.
Overall the manuscript is quite well written, but there are several caveats. As it is, the manuscript is not suitable for publication. Before publication the manuscript should be revised extensively in its content.
The experimental design is good, findings look valid and interesting but overall presentation of findings is not clear to the reader.
Several general comments are listed below:
1. A lot sentences are not referred correctly neither to figures, tables nor previously published papers. Furthermore, authors report results of a molecular dynamics simulation that is not referred neither in the methods nor in another published paper.
2. Figures are not well explained and the correct order of figure citations is not respected.
3. Colored figures on black background are not clear

Experimental design

Overall the experimental design is good. However, authors refer to a molecular dynamics simulation that is not explained at all in the methods. Furthermore, this experiment is not refered to any previous paper.

Validity of the findings

Data are robust and conclusions are well written. I strongly suggest to improve explanation of figures.

Additional comments

Further comments to each section are listed below.

Abstract
Few grammatical errors and typos should be fixed.
The sentence from line 47 and 50 needs to be rephrased

Introduction
Line 105. The in-silico term is not necessary. I suggest “ ….has been built using conventional molecular modeling…”

Results
-Clusterization of RPB1 is evident, however in order to help the reader I suggest to modify Figure 1 and supplementary figure 1 by adding something that might highlight the clusterization (empty circle or a line).
- lines 260-262 control the position of commas.
- line 262 “Good conservation is evident”. Evident from what? Please refer to a figure (figure 2) or a table or a paper.
- line 271 “…have high identity score”. Again, please refer to a figure (Figure 2?) or a table.
- lines 271-273. The authors are right, however, there are several reports about the important role of conserved motifs in protein function. Authors should do the effort to search in literature and find one or two references about.
- Line 293. Here, there should be Figure 3 not Figure 5. Furthermore, there is not match with reference in the text and caption of figure 5.
- Lines 293-300. The authors have not well explained what is described in figure 3. Did they include in the alignment the sequences extracted from the crystal structures? Figure 3 reports only two aligned sequences. Additionally, Figure 3 seems to report references to domains of the crystal structures on the right site of the alignment, this should be described in the text and not only in the Figure 3 caption.
-line 309. Figure 5 is ok; its reference in this part of the text is right and not in line 293. Figure 6 is not necessary and it should be substituted by a table. This putative new table can shows also results from usage of Verify3D. NB. Superimposition should be shown with cartoons of different colors, and not as superimposition of spheres and surfaces.
- Lines 324-325. Rephrase the first sentence and add a reference to the second sentence.
-Lines 341-342. I cannot see the highlight of zinc finger in figure 4
-Lines 345-347. “The second zinc-finger can be recognized…”. From what? Please refer to the right figure.
-Lines 347-349 “Finally, according to our molecular dynamics simulations…”. There is no reference at all to this sentence. When and how the authors have carried out the molecular dynamics simulation? Where those results are reported?

Reviewer 2 ·

Basic reporting

Line 88-90: reference is required in order to support the argument of the authors.

Line 97-98: in my opinion, the reference for the argument here is out of date (since 2001). As the authors mention that the three-dimensional crystal structure remains unknown so far, I suggest a more recent reference to be cited.

Line 103: I could not find the direct link between the cited reference (Samish, at al., 2015) and the argument of the sentence. I suggest that the authors replace the reference here by a relevant one.

Figure 4C and 4D: the notation of the motifs and domains is not clear enough. I suggest that the authors increase the font size of the notation.

Experimental design

No Comments

Validity of the findings

Line 347-349: lacking data of molecular dynamics simulations for the conclusion here. I suggest that the authors provide the root mean square deviation (RMSD) and the root mean square fluctuation (RMSF) of the protein backbone during the simulation to support the conclusion (put in the supplementary).

Additional comments

TBB DdRpII RPB1 protein is an enzyme of substantial interest in the field of TBB infection research. Within this study, the authors propose new sites of high pharmacological interest on the protein and suggest novel pharmacophore models for fast screening of potential inhibitors. The manuscript is logically presented, and uses various state of the art computational techniques in a thorough manner to reach its conclusions. Overall, the study was interesting and informative.

Beside the comments on Basic Reporting and Validity of the Findings above, the only minor recommendation prior to publication is for checking carefully the use of references throughout the manuscript.

·

Basic reporting

The manuscript about the modeling of TB DdRpII is useful of new drug discovery. The research methods and results are standard. The authors conducted homology modeling and performed the MD study. They also validated their models. Finally, the build a pharmacophore for further drug study. The results are interesting and reasonable and the manuscript are organize in good style. I have some comments about the paper:
1. I suggested to revised the title as "A comprehensive phylogenetic and structural study of TB DdRpII towards the designing of novel antiparasitic agents." The importance about the paper is not necessary to demonstrate in the title.
2. the sentence in abstract line 8 should be revised " since it is instrumental role is vital for the parasite’s survival, proliferation, ...."
3. In the method part, in "molecular dynamic", abbreviation about "Root-mean-square deviation" should be consistent in the paper using "RMSD"
4. It is better to include the result of "Model evaluation" in SI.
5. Because you mentioned in method part, it is better to include the result of virtual screening for other researchers' reference.
6. About the Zinc domain, I knew it is very hard to model and I have some considering. The "motif 1a and motif 1b" is not found in figure 4, please label it. Because the Zinc ion is important for the structure. Did you consider the function of Zinc ion for the construction of this model. Can the modeling method well construct the Zinc domain? Did you add the Zinc ion in the MD study? What differences of your model of Zinc finer domain with template.

Experimental design

No Comments

Validity of the findings

No Comments

---

## Round 0.2 · Major Revisions

Like reviewer 1, I am troubled by the lack of hard data from the molecular dynamics simulations. Timecourses of RMSD (vs. initial model structure) for the whole protein and for each of its structural motifs must be shown and discussed in the main text. I am also not completely comfortable with your choice of such a poor template structure (3H0G). I do understand that was the best template available until the end of 2015, but 5FLM (released on January 10, 2016) has almost perfect validation statistics and similar identity (and higher similarity) to T. brucei RNA pol II. You may want to compare the structure of the model derived from this better structure to your model, just to characterize the portions of your model where quality is expected to be worst (or best). I am not demanding you to perform a 100 ns MD using a 5FLM-based model, but to more extensively validate your model in light of this better structure.

Other points:
- The specific identifier Brucei is (wrongly) capitalized throughout the text. Please change Trypanosoma Brucei to Trypanosoma brucei throughout. This also happens in regard to other species (e.g. S. pombe, which is referred to as "S. Pombe", etc.)

- some references do not seem relevant: For example, in "The molecular model has been constructed using conventional molecular modelling techniques and a known 3D crystal structure of Schizosaccharomyces Pombe RNA Polymerase II as a template (Papageorgiou et al. 2014b)" , the quoted reference does not describe any crystal structure from S. pombe proteins, nor is it a landmark paper on molecular modelling techniques. Multiple alignment with ClustalW is referred to "CLUSTALW (Papageorgiou et al. 2016; Thompson et al. 1994) statistical method" , where (as far as I can tell) Papageorgiou et al. 2016 does not refer to any modifications of the CLUSTALW algorithm.

Reviewer 1 ·

Basic reporting

The authors have improved the manuscript, however there are few things that should be fixed. My main concerns are about the molecular dynamics study.
I still think that figure 5 needs some improvement.
I suggest to do not use the spheres representation for each superimposed structure, instead superimpose structures with cartoon or ribbon representations.

Experimental design

The authors have carried out 100 ns of molecular dynamics simulation, however they do not explain outcomes from MD simulations in the results session. The authors report only the plot of energy vs time. Parameters of structure flexibility are generally the RMSD and RMSF. The authors report in the discussion "according to our molecular dynamics simulations, the main role of the Rpb1 and Rpb2 subunits is to provide stability within the overall structure formation of the RNA polymerase II molecule in the 3D space.". The plot of the energy of the system is not enough to support the sentence above.

Validity of the findings

The finding reported in the discussion, but not in results, "according to our molecular dynamics simulations, the main role of the Rpb1 and Rpb2 subunits is to provide stability within the overall structure formation of the RNA polymerase II molecule in the 3D space" is not supported by the figure about molecular dynamics simulation. In fact energy is generally conserved after the pre-equilibration state, and parameters used for explain protein stability and flexibility are others. I suggest the authors to report RMSD of whole protein, RMSD of each motif; along to RMSF data (protein and motifs). If these data do not explain about the flexibility of protein the molecular dynamics studies are totally meaningless, thus I suggest to report only the RMSD of superimposition of model before and after MD.

---

## Round 0.3 · Minor Revisions

The manuscript has been greatly improved. A few cosmetic changes are, however, still needed: the numbers in the axes of several figures are extremely small, and the precise boundaries of domains A-E are not easy to find. Would you mind adding those data to the legends of figure 8. I agree that the figures 9-11 may be moved to the Supporting Information section, but the size of their axes' labels should be increased.

---

## Round 0.4 · accepted · Accept

There are still a few cosmetic issues than must be corrected, but I believe you may address them directly with our production staff:
line 327 : "full coarse" should be "full course"
The axes labels in Figure 7 should be changed to a larger font. "Amstrong" should be changed to "Angstrom"